# Antibody-dependent enhancement of toxicity of myotoxin II from *Bothrops asper*

Christoffer V. Sørensen[1], Julián Fernández[2], Anna Christina Adams[3], Helen H. K. Wildenauer[1], Sanne Schoffelen[1], Line Ledsgaard[1], Manuela B. Pucca[4], Michael Fiebig[5], Felipe A. Cerni[6], Tulika Tulika[1], Bjørn G. Voldborg[1], Aneesh Karatt-Vellatt[7], J. Preben Morth [1], Anne Ljungars [1], Lise M. Grav[1], Bruno Lomonte [2] ✉ & Andreas H. Laustsen [1] ✉

Improved therapies are needed against snakebite envenoming, which kills and permanently disables thousands of people each year. Recently developed neutralizing monoclonal antibodies against several snake toxins have shown promise in preclinical rodent models. Here, we use phage display technology to discover a human monoclonal antibody and show that this antibody causes antibody-dependent enhancement of toxicity (ADET) of myotoxin II from the venomous pit viper, *Bothrops asper*, in a mouse model of envenoming that mimics a snakebite. While clinical ADET related to snake venom has not yet been reported in humans, this report of ADET of a toxin from the animal kingdom highlights the necessity of assessing even well-known antibody formats in representative preclinical models to evaluate their therapeutic utility against toxins or venoms. This is essential to avoid potential deleterious effects as exemplified in the present study.

Snakebite envenoming is a devastating neglected tropical disease that kills and permanently disables hundreds of thousands of people each year[1–3]. In Central America and northern South America, the venomous pit viper *Bothrops asper* is known for causing a large number of bites that result in mortality and morbidity[4]. The venom of *B. asper* contains a significant amount of potent myotoxic phospholipases A$_2$ (PLA$_2$s) and PLA$_2$-like proteins, which are responsible for serious clinical effects, such as skeletal muscle necrosis, sometimes leading to amputation and permanent disability[5–7]. One of the important toxins from *B. asper* is myotoxin II, which is classified within the Lys49 PLA$_2$-like toxin category, characterized by the substitution of the catalytically vital Asp49 residue with lysine[8]. This modification results in their enzymatic inactivity, while preserving their robust myotoxic activity. Although the precise mechanism of action for Lys49 PLA$_2$-like toxins remains only partially understood, studies propose that a stretch of

cationic and hydrophobic amino acids at positions 115–129 constitute the toxic site of myotoxin II. The toxic mechanism of Lys49 proteins has been associated with a dimeric toxin state; however, recent analyses have shown that myotoxin II exists in a monomeric state under native conditions[9].

PLA$_2$s and PLA$_2$-like myotoxins are known to act locally, binding to the plasma membrane of skeletal muscle fibers and affecting its integrity to induce necrosis[6,10]. The binding of myotoxins to muscle fibers at the vicinity of the injection site essentially precludes these toxins from diffusing into the lymphatics and reaching circulation[11,12]. Myotoxin II is considered a key toxin to be neutralized for treatments to be efficacious in minimizing local tissue damage in patients envenomed by *B. asper*[5,6]. The mainstay treatments against snakebite envenomings are antivenoms derived from the plasma of hyper-immunized animals[13]. While effective, these treatments are also

[1]Department of Biotechnology and Biomedicine, Technical University of Denmark, DK-2800 Kongens Lyngby, Denmark. [2]Instituto Clodomiro Picado, Facultad de Microbiologia, Universidad de Costa Rica, San Jose, Costa Rica. [3]The Novo Nordisk Foundation Center for Biosustainability, Technical University of Denmark, DK-2800 Kongens Lyngby, Denmark. [4]Medical School, Federal University of Roraima, Boa Vista BR-69310-000, Brazil. [5]Absolute Antibody Ltd, Wilton Centre, Redcar, Cleveland TS10 4RF, UK. [6]Postgraduate Program in Tropical Medicine, University of the State of Amazonas, Manaus BR-69040-000, Brazil. [7]IONTAS Ltd., Cambridge, UK. ✉e-mail: bruno.lomonte@ucr.ac.cr; ahola@bio.dtu.dk

associated with several drawbacks, such as the risk of causing adverse reactions[14,15], high costs[16], and low concentrations of therapeutically active antibodies[17]. To avoid these drawbacks, novel types of snakebite therapeutics known as recombinant antivenoms, based on human monoclonal antibodies, are under development. A number of human monoclonal immunoglobulin G (IgG) antibodies have been reported to neutralize snake toxins in vivo, and it has been speculated that this type of antibodies could find broad therapeutic utility in the neutralization of different families of snake toxins[18–21]. However, common for all these previously reported human monoclonal IgG antibodies is that they target neurotoxins, which exert their function extracellularly by interacting specifically with cellular receptors.

In this work, we focused on a different type of toxin, namely the membrane-disrupting toxin myotoxin II from *B. asper*. Using a similar workflow as previously described[18,20–22], we developed a myotoxin II targeting human monoclonal antibody in both a Fab format and in two IgG1 formats differing in their Fc region mutations, one containing the LALA[23] mutation and one containing both the LALA and the YTE[24] mutations. These mutations result in reduced binding to Fc-gamma receptors and half-life extension through increased binding to the neonatal Fc receptor (FcRn), respectively. The three antibody formats were tested in two different rodent models: One model involving preincubation of venom and antibody prior to intramuscular (i.m.) administration (preincubation assay), and the other model involving i.m. administration of venom, followed by intravenous (i.v.) administration of the antibody upon a time delay (rescue assay). While the IgG antibody showed superior neutralizing effects in the preincubation model, we observe a striking phenomenon in the rescue assays, namely that the YTE-mutated IgG antibody enhances the myotoxic effects of *B. asper* venom. Moreover, some mice receiving both the antibody in Fab format and venom died a day after injection, both in the preincubation and rescue assay, despite the Fab being able to neutralize myotoxicity. This curious phenomenon is termed antibody-dependent enhancement of toxicity (ADET) to distinguish it from other cases of antibody-dependent enhancement (ADE), such as those found in virology and vaccinology[25,26]. The ADET phenomenon has never before been reported for a toxin from the animal kingdom, but has, however, been reported for the poisonous mushroom (*Amanita* genus) toxin α-amanitin and Toxin A from *Clostridium difficile*[27,28]. Beyond the ADET discovery, our findings highlight the necessity of assessing even well-known antibody formats in representative preclinical models, such as rescue models mimicking a snakebite envenoming case, to evaluate their therapeutic utility, as well as the need for careful design and engineering of monoclonal antibodies to ensure that they can neutralize toxins without causing ADET.

## Results

### Phage display selection and initial screening of monoclonal scFvs

Using a naïve fully human antibody phage display library[29], single-chain variable fragment (scFv) displaying phages were selected against myotoxin II from *B. asper*. Three consecutive rounds of panning against myotoxin II were carried out to enrich for phages displaying scFvs with specificity to myotoxin II. To assess the success of the selection, the specificity of the scFv-displaying phages after the third selection round was tested and confirmed (Fig. 1a).

Upon confirmation of successful accumulation of myotoxin II binders, DNA encoding the scFvs was amplified from the polyclonal phage outputs, and the genes were subcloned and expressed. Characterization of 276 scFv-producing monoclonal cultures using ELISA yielded 184 scFvs binding to myotoxin II (using an arbitrary cutoff of an ELISA signal of 0.2) (Fig. 1b). These 184 scFvs were screened for their specificity to myotoxin II by including milk proteins, streptavidin, neutravidin, and a PLA₂ from *Naja mossambica* as controls. Results demonstrated that all the 184 scFvs were binding specifically to myotoxin II and not to any of the control antigens (Fig. S1). The genes encoding the variable heavy ($V_H$) and variable light ($V_L$) for all clones were sequenced, revealing 123 scFv sequences with unique CDR regions, of which 59 showed unique $V_H$ CDR3 regions.

### Ranking of scFv binding and affinity assessment of IgG-reformatted clones

To rank the clones based on binding affinity, an expression-normalized capture assay was performed (Fig. S2). The six scFvs displaying the highest binding signals, TPL0039_05_E02 (E02), TPL0039_05_B12 (B12), TPL0039_05_F04 (F04), TPL0039_05_G08 (G08), TPL0039_05_B04 (B04), and TPL0039_05_A03 (A03), were selected for reformatting to a human IgG1 format (containing the LALA and YTE mutations in the Fc region), expressed, purified, and their binding to myotoxin II was confirmed for all six IgGs through ELISA (Fig. 2a). Following confirmation of retained binding ability, the functional affinity (avidity) of the six IgGs was measured using bio-layer interferometry experiments, yielding functional affinities ranging from below 1 pM (no dissociation observed) to 9 nM as seen in Fig. 2b. To test how the functional affinity (avidity) compared to the affinity of the individual binding sites, one clone (E02) was expressed as Fab and its

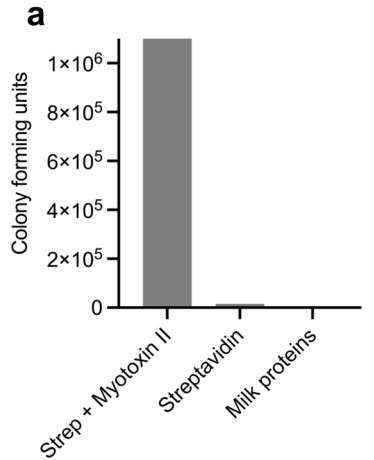
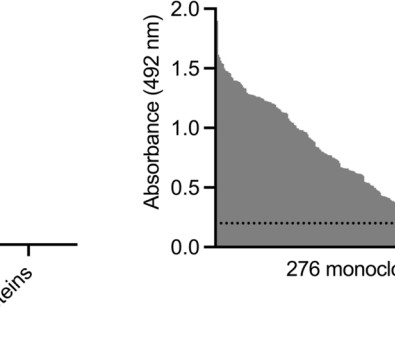

**Fig. 1 | Overview of results from the phage display discovery process.**
**a** Assessment of the specificity of the third round polyclonal phage output reported as colony forming unit (CFU) counts when panned against either myotoxin II or controls (streptavidin and milk proteins). **b** Monoclonal scFv ELISA signals against myotoxin II. The dotted line represents the arbitrary binding cutoff of an ELISA signal of 0.2. Source data are provided as a Source Data file.

**a**

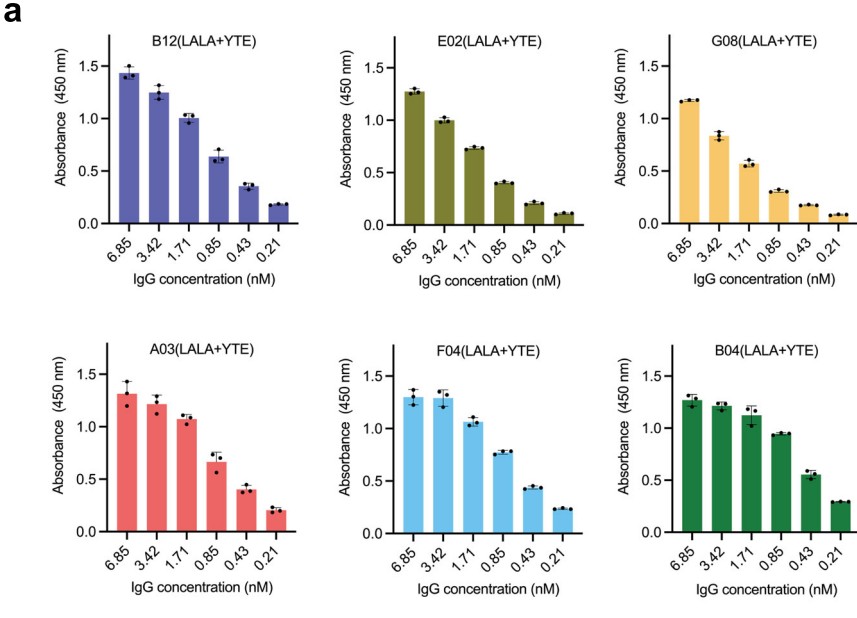

**b**

| Antibody | $K_D$(nM)# | $k_{on}$ (M$^{-1}$s$^{-1}$) | $k_{off}$ (s$^{-1}$) |
|:---:|:---:|:---:|:---:|
| B12(LALA+YTE) | <1.0·10$^{-3}$* | 6.85·10$^5$ | <1.0·10$^{-7}$* |
| E02(LALA+YTE) | <1.0·10$^{-3}$* | 4.91·10$^5$ | <1.0·10$^{-7}$* |
| G08(LALA+YTE) | 9.41 | 3.01·10$^5$ | 2.83·10$^{-3}$ |
| A03(LALA+YTE) | <1.0·10$^{-3}$* | 6.48·10$^5$ | <1.0·10$^{-7}$* |
| F04(LALA+YTE) | 1.81·10$^{-2}$ | 7.23·10$^5$ | 1.31·10$^{-5}$ |
| B04(LALA+YTE) | 7.58·10$^{-2}$ | 8.61·10$^5$ | 6.53·10$^{-5}$ |
| E02(Fab) | 2.42 | 3.60·10$^5$ | 8.72·10$^{-4}$ |

**Fig. 2 | IgG binding assessment. a** Dose-response ELISAs showing binding to myotoxin II of the clones as IgGs. Experiments were performed in triplicates ($n = 3$) and reported as means with error bars representing the SD and dots representing individual data points. **b** Affinity measurements of the six IgGs and one Fab affinity assessed in a bio-layer interferometry experiment. The KD of the E02(Fab) format was measured as 2.4 nM, whereas the IgG version of the clone was measured as <1 pM. determined in bio-layer interferometry experiments. *: The IgG was not observed to dissociate from myotoxin II, thereby resulting in the minimum value. #: Since six of the antibodies were measured as IgGs, the $K_D$ refers to functional affinity (avidity) for these. Source data are provided in a Source Data file.

### Neutralization of cytotoxic effects of myotoxin II using a fibroblast-based assay

To test the neutralization of the cytotoxic effects of myotoxin II by the six IgGs in vitro, a fibroblast viability assay was carried out (Fig. 3). In this assay, three molar ratios of toxin:IgG were tested; 1:0.5, 1:1, and 1:1.5. The included controls were: a negative control (PBS), a positive control (toxin challenge without IgG), an antibody control (0:1.5 toxin:IgG ratio), and a non-myotoxin II binding IgG at toxin:IgG ratios of 1:1.5 and 0:1.5. The negative control showed no influence on cell viability, whereas the positive control resulted in ~80% reduction in viability. The irrelevant IgG displayed no significant change in cell viability when compared to the positive control, suggesting that any neutralization observed in the assay for other IgGs was caused by the specificity to myotoxin II.

The in vitro neutralization experiments involving the six IgGs can be divided into three patterns. In the experiments involving the two IgGs, E02(LALA + YTE) and F04(LALA + YTE), a lower cell viability than for the negative control was observed at all IgG concentrations, with increasing IgG concentrations resulting in decreasing cell viability.

For G08(LALA + YTE) and B04(LALA + YTE), the neutralization capacity was observed to be slightly dose-dependent, with increasing IgG ratios leading to an increase in cell viability and full neutralization (non-significant change from negative control) was observed at 1:1 (G08(LALA + YTE)) and 1:0.5 (B04(LALA + YTE)) toxin:IgG ratios. Finally, B12(LALA + YTE) and A03(LALA + YTE) displayed full neutralization at all tested toxin:IgG ratios, including the lowest ratio with only 1:0.5 toxin:IgG.

### IgG provides full in vivo neutralization of myotoxin II toxicity at low dose in preincubation assays

From the shown data obtained for the six IgGs, we selected B12(LALA + YTE), E02(LALA + YTE), and G08(LALA + YTE) (based on having different binding and in vitro neutralization patterns) for investigation of in vivo neutralization of myotoxin II (Fig. 4). Mice were injected i.m. with preincubated mixtures of myotoxin II and IgG in a 1:1 molar ratio. Control mice were injected with either PBS, myotoxin II, myotoxin II and ICP polyvalent equine antivenom, or myotoxin II and a non-myotoxin II binding IgG. After three hours, the mice were bled, and the plasma creatine kinase (CK) levels were measured as an indicator of muscle damage. Injection of myotoxin II caused a large increase in CK activity levels compared to the PBS-injected control group. When tested, the currently used antivenom for *B. asper* bites,

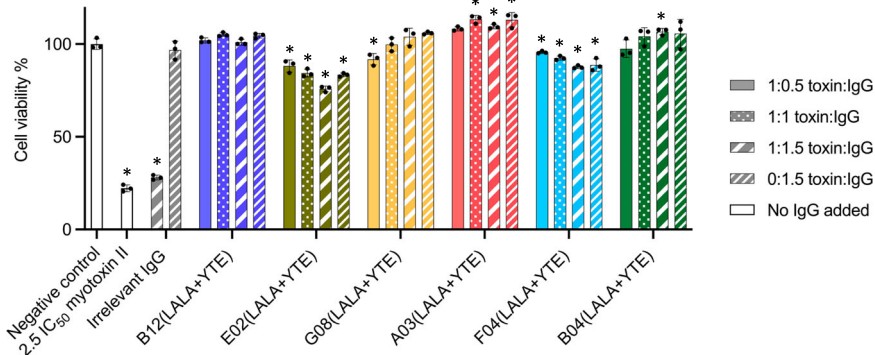

**Fig. 3 | In vitro neutralization of myotoxin II induced cell cytotoxicity by six different IgGs.** Six IgGs were tested at three toxin:IgG ratios to check for myotoxin II (72.5 μg per well) neutralizing abilities in an in vitro fibroblast viability assay. Assay controls included a negative control (PBS), a positive control (toxin challenge without IgG), an irrelevant IgG (a non-myotoxin II binding IgG) at toxin:IgG ratios of 1:1.5 and 0:1.5, and an IgG only control for each tested IgG (0:1.5 toxin:IgG ratio). Experiments were performed in triplicates ($n = 3$) and reported as means normalized to the negative control. Asterisks (*) note statistical difference ($p < 0.05$) compared to the negative control (PBS), error bars represent the SD and dots represent individual data points. Statistics were carried out by comparing each condition to the negative control (PBS) using a one-way ANOVA, followed by Dunnett's multiple comparison test. Source data are provided in a Source Data file, which includes exact $p$-values and confidence intervals.

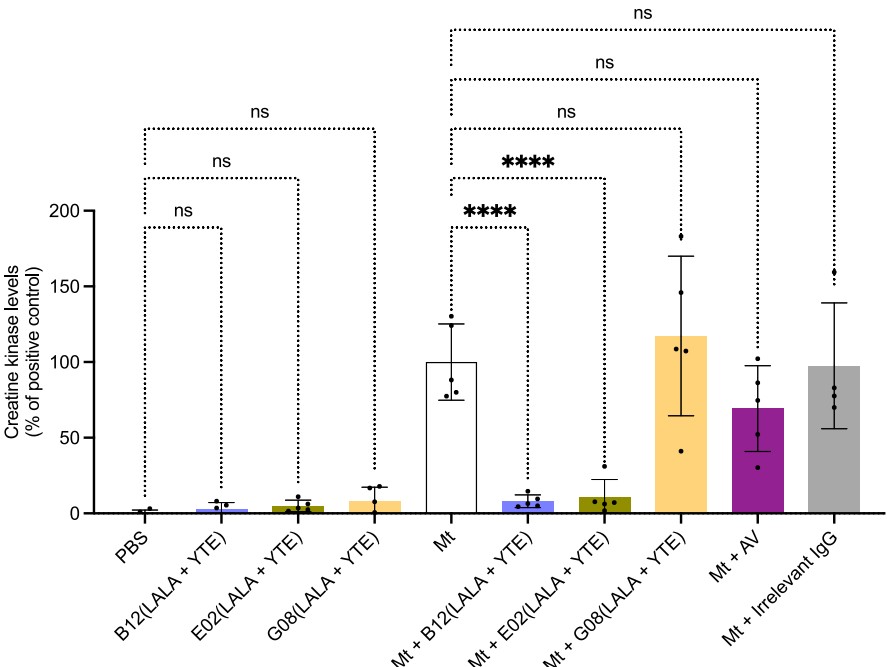

**Fig. 4 | In vivo neutralization of myotoxin II using a preincubation assay.** The graph illustrates in vivo muscle damage caused by myotoxin II (Mt) from *B. asper* measured as increased creatine kinase (CK) levels. Myotoxin II was preincubated with antibody/antivenom for 30 min and then injected i.m. in mice. After 3 h, the mice were bled, and CK levels were measured. To accommodate for assays carried out on different days, the CK levels have been normalized to the positive control (Mt) and are reported as means with error bars representing the SD ($n = 5$ mice, except for Mt + Irrelevant IgG where instead $n = 4$ mice due to the removal of an outlier) and dots representing individual data points. Asterisks (****$p$-value < 0.0001) note significant statistical difference, ns represents a non-significant $p$-value (>0.05). Statistics were carried out on preselected groups using one-way ANOVA followed by Šídák's multiple comparison test. For clearer visualization, the ICP antivenom has been abbreviated as AV. Irrelevant IgG refers to an IgG not binding to myotoxin II. Source data are provided in a Source Data file which includes exact $p$-values and confidence intervals.

ICP polyvalent equine antivenom (AV), showed a neutralizing trend, although the CK level increase was still larger compared to the levels observed for the negative control. G08(LALA + YTE) failed to inhibit the CK increment induced by myotoxin II at a 1:1 toxin:IgG ratio, which could indicate that the antibody may either bind to a non-neutralizing epitope or require a higher toxin:IgG ratio due to its lower affinity. On the other hand, both B12(LALA + YTE) and E02(LALA + YTE) were able to fully prevent the CK level increase caused by myotoxin II at a 1:1 toxin:IgG molar ratio.

**A combination of B12(LALA + YTE) monoclonal IgG and ICP polyvalent equine antivenom neutralizes muscle damage caused by *B. asper* whole venom in preincubation assays**
Additionally, we wanted to assess whether our discovered human IgGs could be used in combination with the ICP polyvalent equine antivenom in vivo to reinforce its neutralizing ability. To assess this, we tested if the ICP antivenom supplemented with B12(LALA + YTE) could neutralize the muscle damage caused by *B. asper* whole venom. In this experiment, mice were injected i.m. with preincubated mixtures of *B.*

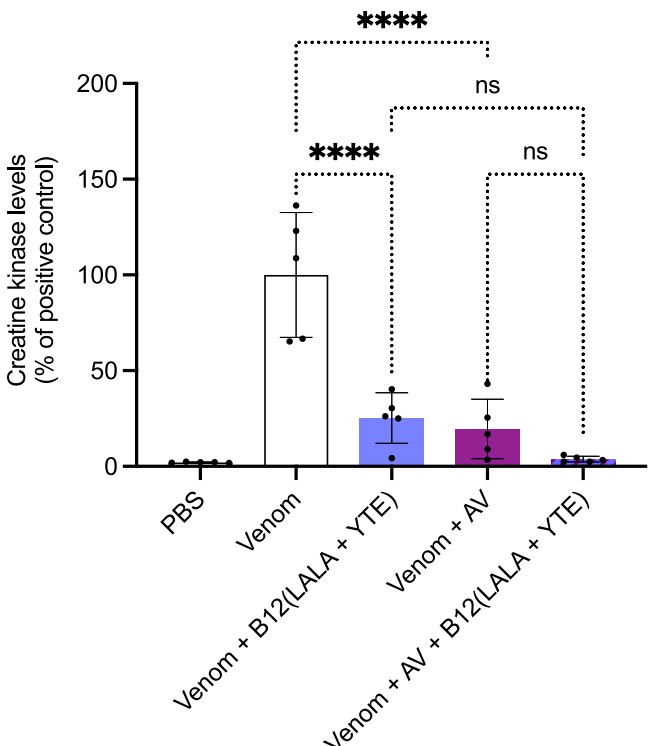

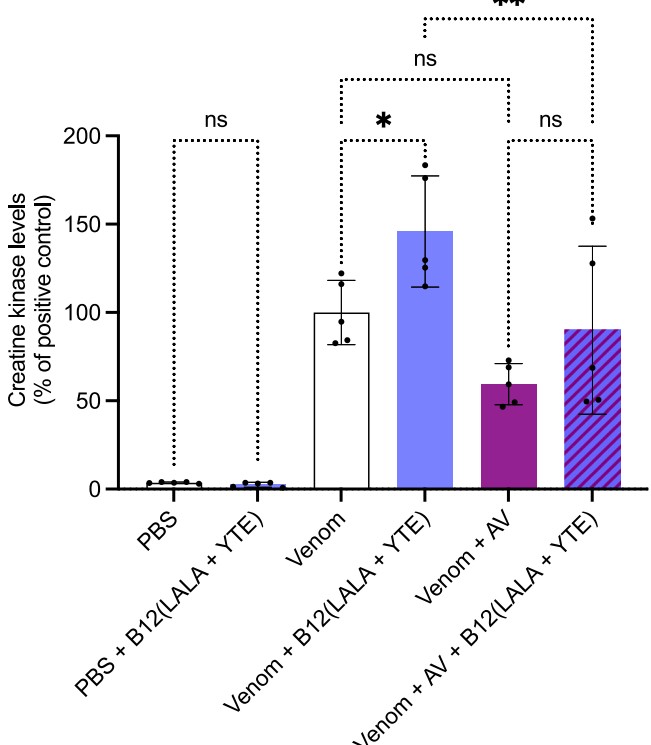

**Fig. 5 | In vivo neutralization of the muscle damaging effects caused by *B. asper* venom using B12(LALA+YTE) in a preincubation assay.** The graph illustrates in vivo muscle damage caused by *B. asper* venom measured as increased creatine kinase (CK) levels. *B. asper* whole venom (Venom) was preincubated with antibody/ antivenom (B12/AV) for 30 min and then injected i.m. in mice. After 3 h, the mice were bled, and CK levels were measured. The CK levels have been normalized to the positive control (Venom) for graph consistency and are reported as means with error bars representing the SD ($n = 5$ mice) and dots representing individual data points. Asterisks (****$p$-value < 0.0001) note significant statistical difference, ns represents a non-significant $p$-value (>0.05). Statistics were carried out on preselected groups using one-way ANOVA followed by Šídák's multiple comparison test. Source data are provided in a Source Data file which includes exact $p$-values and confidence intervals.

**Fig. 6 | In vivo rescue assays using B12(LALA + YTE).** The graph illustrates in vivo muscle damage caused by *B. asper* venom measured as increased creatine kinase (CK) levels. *B. asper* whole venom (Venom) was injected i.m. and 3 min later antibody/antivenom (B12(LALA + YTE)/AV) or PBS was injected i.v. in mice. After 3 h, the mice were bled, and CK levels were measured. The CK levels have been normalized to the positive control (Venom) for graph consistency and are reported as means with error bars representing the SD ($n = 5$ mice) and dots representing individual data points. Asterisks (*$p$-value < 0.05, **$p$-value < 0.01) note significant statistical difference, ns represents a non-significant $p$-value (>0.05). Statistics were carried out on preselected groups using one-way ANOVA followed by Šídák's multiple comparison test. Source data are provided in a Source Data file which includes exact $p$-values and confidence intervals.

*asper* whole venom and ICP antivenom supplemented with B12(LALA + YTE). Control mice were injected with either PBS, *B. asper* venom, *B. asper* venom and ICP polyvalent equine antivenom, or *B. asper* venom and B12(LALA + YTE). Injection of *B. asper* venom caused a large increase in CK activity levels compared to the PBS-injected control group (Fig. 5). Furthermore, the venom preincubated with B12(LALA + YTE) and venom preincubated with ICP antivenom controls showed partial reduction of the CK levels. When tested together against *B. asper* venom, ICP antivenom and B12(LALA + YTE) showed CK levels almost identical to the PBS only control (although the statistical difference from venom + B12(LALA + YTE) and venom + antivenom is not significant), indicating full neutralization of the muscle-damaging effects caused by the whole venom.

### The B12(LALA + YTE) antibody switches from toxin-neutralizing to toxin-enhancing when tested in rescue assays instead of preincubation assays

Next, we tested the neutralizing capabilities of B12(LALA + YTE) in a rescue assay since it more closely resembles a real-life envenoming. As a positive control, injection of venom alone was used which resulted in an increase in CK levels, and the negative control (PBS) resulted in no increase in CK levels (Fig. 6). Injection of venom followed by injection of B12(LALA + YTE) resulted in a significant increase in CK levels compared to the positive control. The ICP polyvalent antivenom decreased the CK level compared to the venom only, although, while

not statistically significant, the CK levels appeared to increase when the antivenom was combined with B12(LALA + YTE) compared to when the ICP polyvalent antivenom was used alone to treat the envenomed mice. This increase in CK levels when treating the envenomed mice with B12(LALA + YTE) was thus observed both when the monoclonal antibody was used alone and in combination with the ICP polyvalent antivenom, however, when tested without venom, B12(LALA + YTE) caused no increase in CK levels.

The unexpected switch from toxin-neutralizing to toxin-enhancing could potentially be explained by morphological alterations observed in the kidneys (Fig. 7). Here, it seems that the envenomed mice treated with B12(LALA + YTE) suffer from increased kidney damage compared to envenomed mice without treatment. The examination of the Bowman's capsules led to the observation that these capsules appear to be substantially compromised in envenomed mice treated with B12(LALA + YTE) (Fig. 7g, j) compared to untreated envenomed mice (Fig. 7a, b, and d). An additional morphological difference observed is the increased erythrocyte dispersal in envenomed mice treated with B12(LALA + YTE) (Fig. 7i, k, and l) compared to similar tissues in untreated envenomed mice (Fig. 7c, e, and f). Histological observations in lung, liver, and heart tissue revealed no apparent difference between B12(LALA + YTE) treated or untreated envenomed mice (Fig. S3).

To further investigate this switch in antibody function we modified the mutations of the B12 IgG to only contain the LALA mutation,

Venom only

Venom + B12(LALA+YTE)

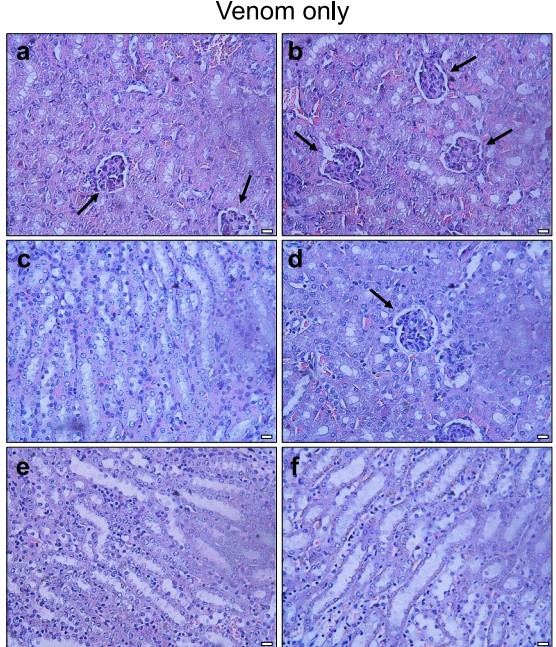
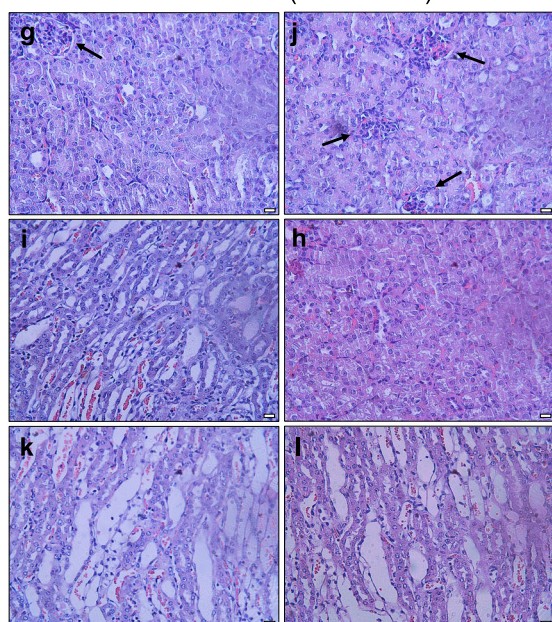

**Fig. 7 | Histological examination of kidney tissues from envenomed mice either untreated or treated with B12(LALA + YTE) in a rescue assay. a–f** Venom only, (**g–l**) Venom + B12(LALA + YTE). Black arrows are pointing at intact and compromised Bowman's capsules. The scale bar in the bottom right corner of each picture is equal to 20 μm. Results from one independent experiment are shown. The experiment was carried out with two mice in each group with consistent results between the duplicates.

instead of both the LALA and YTE mutations as in earlier experiments. This change in the IgG scaffold appeared to remove the CK level increase that the B12 IgG caused when administered after venom injection, either in combination with the ICP polyvalent antivenom or alone (Fig. 8). Further, when testing the B12 IgG without the YTE scaffold mutation, no mice exhibited behavioral changes and no mice died during the extended observation.

### In the Fab format, B12 shows significant neutralization of myotoxicity in rescue in vivo assays

Lastly, an assessment of the B12 antibody in Fab format was performed using a preincubation in vivo assay, demonstrating that B12(Fab) was able to significantly decrease the CK levels (Fig. 9). It was furthermore shown, that B12(Fab) could significantly reduce the CK levels in a rescue assay (Fig. 9). To compare with the IgG neutralization experiments, the mice were kept for an extended period, after which 1/5 mice died in both the preincubation and the rescue assay.

## Discussion

In this study, high affinity fully human recombinant monoclonal IgG antibodies targeting myotoxin II from *B. asper* venom were discovered using phage display technology directly from a naïve antibody library. A subset of these antibodies was able to neutralize the cytolytic effect of myotoxin II on human fibroblasts in vitro. In mice, two of these antibodies (B12(LALA + YTE) and E02(LALA + YTE)) completely neutralized the muscle-damaging activity of myotoxin II, measured as an increase in plasma CK levels, when tested by preincubation at a 1:1 toxin:antibody molar ratio, followed by i.m. administration of the mixture. Additionally, in this preincubation assay, B12(LALA + YTE) was found to inhibit muscle damage (measured as an increase in plasma CK levels) caused by *B. asper* whole venom, implying its ability to neutralize the main myotoxin iso-form(s) present in this venom[6]. In agreement, the addition of a low proportion (7% by protein content) of B12(LALA + YTE) to an equine polyvalent antivenom showed improved neutralizing trends against the myotoxic effect of *B. asper* whole venom, thereby indicating that monoclonal antibodies are likely compatible with being mixed into existing antivenom products as fortification agents, as has previously been hypothesized[30].

In contrast to the results obtained in the preincubation tests, when employing a rescue model (antibody administered i.v. 3 min after i.m. venom injection), B12(LALA + YTE) unexpectedly caused a significant increase in plasma CK activity, indicating higher muscle damage, in comparison to mice that only received a venom injection. Not only were plasma CK levels increased by B12(LALA + YTE) treatment of envenomed mice, but these mice also became lethargic a few hours after injection and died within 24 h. Injection of B12(LALA + YTE) alone did not cause any significant increments in plasma CK levels or signs of toxicity. Thus, the effect only occurred when this antibody was administered by the i.v. route subsequent to i.m. venom injection. Similar unexpected ADET phenomena have earlier been reported for two other toxins, namely Toxin A from *Clostridium difficile*[27] and the poisonous mushroom toxin α-amanitin, the latter of which showed an increase in toxicity of 2-fold and 50-fold when mice challenged with α-amanitin were treated with anti-α-amanitin IgG and Fab, respectively[28]. In comparison, the mechanism of ADE is also known from other fields, such as vaccinology, where it is connected with viral enhancement[25,26]. Recently, ADE was identified as the underlying cause of breakthrough severe dengue disease in children who had not been previously infected with dengue but had received a yellow fever chimeric tetravalent dengue vaccine[31]. While an entirely different mechanism must be at play in ADET compared to ADE of viruses (given the fundamental differences between a toxic protein and a replicating entity such as a virus), the above examples highlight the importance of investigating and understanding ADET and ADE, as these effects can be detrimental to the development of new therapeutics. In this relation, it is important to note, though, that ADET observed in in vivo models might not translate into clinically significant ADET in humans, and might also differ based on the in vivo model or the dosing regimen used, as has been observed with vaccine-associated enhanced respiratory disease in relation to COVID-19 vaccines[32].

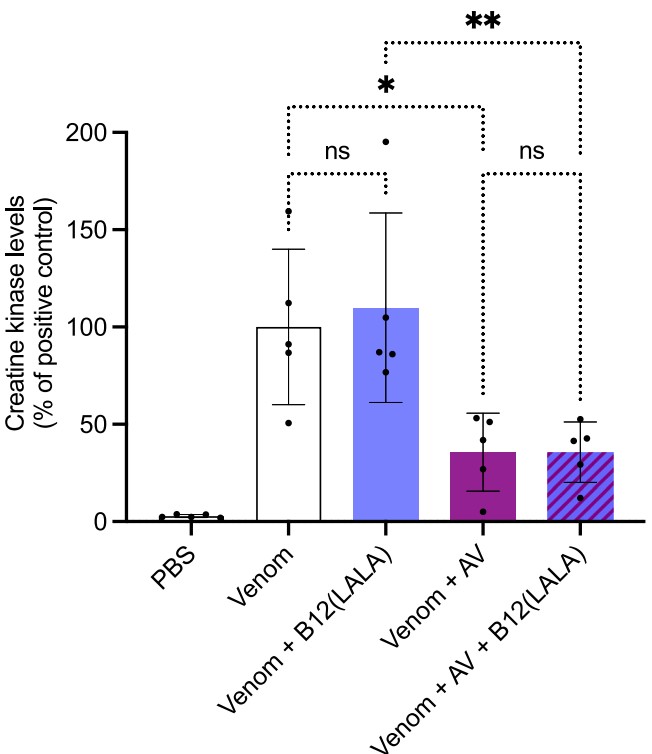

**Fig. 8 | In vivo rescue assays using a different IgG scaffold.** The graph illustrates in vivo muscle damage caused by *B. asper* venom measured as increased creatine kinase (CK) levels. *B. asper* whole venom (Venom) was injected i.m. and 3 min later antibody/antivenom (B12(LALA)/AV) or PBS was injected i.v. in mice. After 3 h, the mice were bled, and CK levels were measured. The CK levels have been normalized to the positive control (Venom) for graph consistency and are reported as means with error bars representing the SD ($n = 5$ mice) and dots representing individual data points. Asterisks (*$p$-value < 0.05, **$p$-value < 0.01) note significant statistical difference, ns represents a non-significant $p$-value (>0.05) Statistics were carried out on preselected groups using one-way ANOVA followed by Šídák's multiple comparison test. Source data are provided in a Source Data file which includes exact $p$-values and confidence intervals.

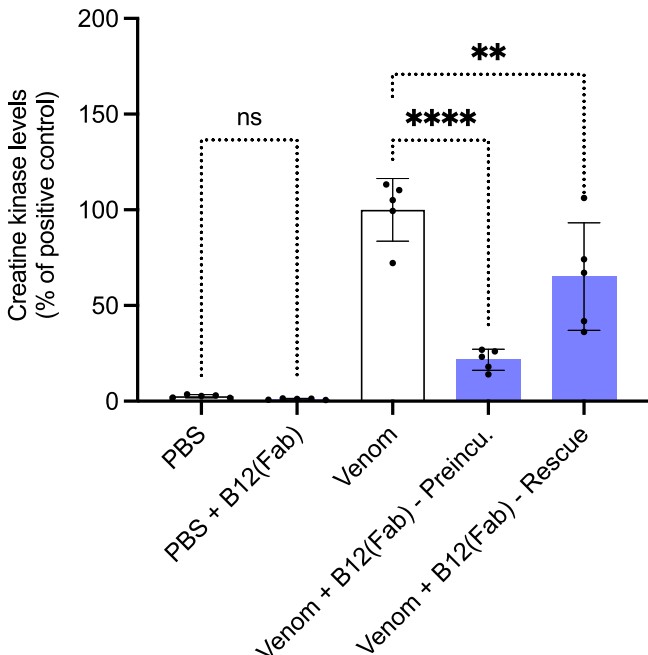

**Fig. 9 | In vivo preincubation and rescue assays using Fabs.** The graph illustrates in vivo muscle damage caused by *B. asper* venom measured as increased creatine kinase (CK) levels. *B. asper* whole venom (Venom) was either preincubated for 30 min with B12(Fab) and injected i.m. or injected without preincubation i.m. followed by i.v. administration of B12(Fab) 3 min later in mice. As the first negative control, PBS alone was injected i.m., and as the second negative control, B12(Fab) alone was injected i.v. After 3 h, the mice were bled, and CK levels were measured. The CK levels have been normalized to the positive control (Venom) for graph consistency and are reported as means with error bars representing the SD ($n = 5$ mice) and dots representing individual data points. Asterisks (**$p$-value < 0.01, ****$p$-value < 0.0001) note significant statistical difference, ns represents a non-significant $p$-value (>0.05). Statistics were carried out on preselected groups using one-way ANOVA followed by Šídák's multiple comparison test. Source data are provided in a Source Data file which includes exact $p$-values and confidence intervals.

Aiming to explore if the observed ADET could be related to the recyclability of the IgG format of the B12(LALA + YTE) antibody, whose scaffold contained the half-life extending mutation YTE[33], further experiments were performed using the B12 antibody with an IgG scaffold without the YTE mutation (B12(LALA)). In addition, the B12 antibody was produced in a monovalent Fab format (B12(Fab)), lacking the entire Fc region. Interestingly, envenomed mice treated with B12(LALA), in contrast to those treated with B12(LALA + YTE), did not show behavioral alterations or deaths, and did not alter plasma CK levels, as compared to mice injected with venom alone. These results indicate that the toxicity could be related to an increased half-life of the antibody-toxin complex or to increased FcRn-mediated uptake of the antibody-toxin complex. When tested in the Fab format, the B12 antibody demonstrated significant neutralization of myotoxicity in both the preincubation and rescue in vivo assays, while simultaneously causing the death of one out of five mice in each assay. Thus, the B12(Fab) experiment indicates that the mechanism responsible for the ADET effect might also involve other pharmacokinetic properties than the YTE-related increased half-life of the B12 monoclonal antibody. Earlier studies on ADET have demonstrated that various mechanisms can be at play. As a first example, it has been shown that mice poisoned with the mushroom toxin α-amanitin suffer from nephrotoxicity rather than hepatotoxicity when treated with an IgG or a Fab antibody. This change in toxicity is attributed to the α-amanitin-antibody complexes being subject to glomerular filtration and tubular reabsorption, unlike

α-amanitin alone[28]. As another example of ADE, it has been shown that during a secondary infection with a dengue virus serotype that is different from the previous infection, the risk of severe dengue fever is increased. This heightened risk is attributed to ADE of infection, whereby sub-neutralizing antibodies bind to virus particles, facilitating their entry into cells through the formation of immune complexes that interact with fragment crystallizable gamma receptors (FcγR) found on monocytes, dendritic cells, and macrophages[34]. As a final example, ADE has also been reported for infections with *Streptococcus pneumoniae*, where bacterium-specific antibody responses enhance bacterial attachment to respiratory cells[35,36]. With these examples in mind and inspired particularly by the methods previously used to study α-amanitin-binding antibodies in Faulstich et al.[28], we carried out histological observations of the kidney, liver, lung, and heart sections from the mice that died after B12(LALA + YTE) treatment and compared these to mice injected with venom alone (Figs. 7 and S3). In these histological observations, we found indications that the integrity of the Bowman's capsules was compromised in envenomed mice treated with B12(LALA + YTE) (Fig. 7g and j) compared to untreated envenomed mice (Fig. 7a, b, and d). This could suggest that a similar mechanism as described in Faulstich et al.[28] is at play in this observed case of ADET, i.e., that the antibody-myotoxin II complexes are subject to glomerular filtration and tubular reabsorption, which myotoxin II might not otherwise be.

Further studies are needed to fully elucidate the complex underlying pharmacology of the ADET effect described in this study.

However, regardless of the nature of the mechanisms at play in the rodent models employed here, our findings highlight a previously unidentified limitation of preincubation assays, which are currently used as the gold standard for assessing the preclinical efficacy of toxin-neutralizing agents in the field of antivenom research[37]. It seems evident that while the preincubation model may still be highly useful, it cannot alone be used to evaluate new types of antivenom products, such as those based on human monoclonal antibodies[38]. Furthermore, our results may even question whether ADET could also occur when existing plasma-derived antivenoms are used and whether this effect may have been entirely overlooked in the past. With this in mind, we recommend antivenom researchers to carefully consider the utility of their preclinical model(s) for evaluating the potential of new, and possibly existing, therapeutic agents against snake toxins and venoms, and possibly include more advanced models, such as rescue assays, to better assess preclinical efficacy[38,39]. Thereby, the elimination of therapeutic leads with undesired effects can potentially be expedited in the drug discovery process, consequently facilitating the development of life-saving next-generation snakebite therapeutics.

## Methods

Protocols for in vivo experiments were approved by the Institutional Committee for the Use and Care of Animals (CICUA), University of Costa Rica (approval number CICUA 84–17).

### Purification of myotoxin II

Myotoxin II (Uniprot P24605) was purified from the venom of *Bothrops asper* by cation-exchange chromatography on CM-Sephadex C25, followed by reverse-phase HPLC on $C_{18}$[7,8]. In brief, for cation-exchange on CM-Sephadex C25, venom was dissolved in 0.1 M Tris, 0.1 M KCl (pH 7.0) and applied to the CM column (20 × 2 cm) equilibrated in the same buffer, and monitored at 280 nm. After elution of unbound proteins at 0.4 mL/min, a linear gradient towards 0.1 M Tris, 0.75 M KCl (pH 7.0) was applied. The last eluting peak was collected, and aliquots of this fraction were then applied to a semi-preparative RP-HPLC C8 column (250 × 10 mm), equilibrated with solution A (0.1% trifluoroacetic acid in water). Elution was performed at 2.5 mL/min, monitored at 215 nm, using a linear gradient towards solution B (0.1% trifluoroacetic acid in acetonitrile). Myotoxin-II was collected, lyophilized, and stored at −20 °C until use. Protein identity was confirmed by determining the deconvoluted monoisotopic intact mass to match with its amino acid sequence (P24605) by direct infusion in a Q-Exactive Plus (Thermo) mass spectrometer.

### Biotinylation of myotoxin II

Lyophilized myotoxin II was solubilized in phosphate-buffered saline (PBS) and mixed with biotin linked to N-hydroxysuccinide (NHS) via a $PEG_4$ linker (EZ-Link™ NHS-PEG₄-Biotin, Thermo Scientific, #A39259) in a molar ratio of 1:1.5 (toxin:biotin) and incubated at room temperature for 30 min. For purification of the biotinylated myotoxin II, buffer exchange columns (Vivacon 500, Sartorius, 3000 Da Molecular Weight Cut-Off) were used following the manufacturer's protocol. The final concentration of biotinylated myotoxin II was measured at 280 nM using a Nanodrop Spectrophotometer (NanoDrop One$^C$ Spectrophotometer, Thermo Scientific).

### Phage display selection and assessment of polyclonal output

For phage display selection, the IONTAS phage display library λ was employed, which is a naïve human antibody phage display library with a clonal diversity of $1.6 \times 10^{10}$. The displayed antibodies come in the form of scFvs and have been constructed from B lymphocytes collected from 43 non-immunized human donors[29].

Phage display selection was carried out as earlier described[29], with the following modifications described briefly: For selections, biotinylated myotoxin II (10 μg/mL) was captured on streptavidin-coated

(10 μg/mL) Maxisorp vials. In the second and third selection round, neutravidin was used instead of streptavidin to prevent accumulation of streptavidin-recognizing antibodies. Elution in the third round was carried out by incubating with a 10 mM glycine-HCL solution at pH 6 for 15 min instead of eluting with trypsin.

After three rounds of phage display selection, antigen specificity of the phage output was evaluated. This was carried out according to Føns et al.[40], using a similar protocol to the phage display selections. The phage output was purified utilizing polyethylene glycol precipitation[29], and binding was tested against Maxisorp vials coated with either (i) biotinylated myotoxin II indirectly immobilized using streptavidin, (ii) streptavidin, or (iii) 3% (w/v) skimmed milk in PBS (M-PBS). Biotinylated myotoxin II and streptavidin were used at a concentration of 10 μg/mL.

### Sub-cloning, primary screening, and sequencing of scFvs

To express soluble scFvs, the scFv genes from the third selection round were sub-cloned from the phage display vector (pIONTAS1) into the pSANG10-3F expression vector using NcoI and NotI restriction endonucleases. The expression vectors were then transformed into *E. coli* BL21(DE3) cells (New England Biolabs), following protocols from Martin et al.[41]. Individual scFv-producing monoclonal colonies (276 colonies) were picked and expressed in 96-well format using auto-induction media[42]. To determine the binding of the scFvs to myotoxin II, a monoclonal scFv ELISA was carried out. 96-well Maxisorp plates with biotinylated myotoxin II (5 μg/mL) captured on streptavidin (10 μg/mL) were used. For detection of binding, a 1:20,000 dilution of anti-FLAG M2-Peroxidase (HRP) (Sigma Aldrich, #A8592) antibody in 3% M-PBS was used followed by the addition of an OPD solution (Sigma-Aldrich, P5412) as substrate according to the manufacturer's protocol. A biotinylated RP-HPLC purified fraction from *Naja mossambica* venom containing $PLA_2$ was included as a negative control antigen. Additional controls included wells coated with streptavidin/neutravidin (10 μg/mL) and 3% M-PBS. Binding was measured as absorbance at 492 nm using an Epoch spectrophotometer from Biotek (15020518) with the Gen5 2.07 software.

The genes encoding scFvs binding specifically to myotoxin II with absorbance (492 nm) values above a pre-set threshold of 0.2 were sequenced (Eurofins and Macrogen genomics sequencing service) using the S10b primer (GGCTTTGTTAGCAGCCGGATCTCA). The antibody framework and CDR regions were annotated and analysed using the Jalview 2.10.5 software and Abysis.org program to identify unique scFvs.

### Expression-Normalized Capture (ENC) DELFIA

Black MaxiSorp plates (Nunc) were coated overnight with anti-FLAG M2 antibody (Sigma, F1804 − 2.5 μg/mL in PBS) at 4 °C. Following blocking, 123 individual supernatants[42] containing monoclonal FLAG-tagged scFvs in 3% M-PBS were added. Next, antigens were introduced at a concentration of 10 nM followed by detection of binding using europium-labeled streptavidin (PerkinElmer 1244−360, 200 ng/mL) in DELFIA assay buffer (PerkinElmer 4002−0010), and DELFIA enhancement solution (PerkinElmer 4001−0010). Binding was measured as Time-Resolved Fluorescence (TRF) using a FLUOstar Omega reader with excitation and emission wavelengths of 337 nm and 615 nm, respectively, and 400 μs as both integration start and integration time.

### Conversion from scFv to antigen-binding fragment (Fab) format

The $V_L$ and $V_H$ sequences from E02 were cloned into the pINT12 expression vector containing the respective heavy chain constant domains and light chain constant domain. The individual variable domains were PCR amplified from the pSANG10-3F vector using LLINK2_F (CTCTGGCGGTGGCGCTAGC) and 2097_R (GATGGTGATG ATGATGTGCGGATGCG) for the $V_L$, and pSang10_pelB (CGCTGCCCA

GCCGGCCATGG) and HLINK3_R (CTGAACCGCCTCCACCACTCGA) for the V$_H$. The PCR amplicons were prepared by digestion with *Nhe*I and *Not*I (V$_L$ digestion) and *Nco*I and *Xho*I (V$_H$ digestion) endonucleases. A four-part ligation, including both variable domains, the pINT12 vector containing the respective heavy chain constant domains, and a stuffer region containing the C$_L$ and CMV promoter cut with *Nco*I and *Not*I, was performed with T4 DNA ligase (Roche, 10481220001).

For B12, the variable chains (V$_L$ and V$_H$) were amplified from the pSANG10-3F vector by PCR and cloned into a single expression vector using the NEBuilder® cloning technique. The expression vector contained the constant heavy domain 1 and constant light domain sequences of human IgG1. The constant heavy domain 1 (ending on VEPKSC) was extended with three alanine residues and six histidine residues for purification purposes. Following transformation and sequence verification the plasmids were used for production.

### Production of Fabs
The E02 Fab was produced by transient mammalian expression using a ratio of 1 µg DNA/mL Expi293F™ cells (Life Technologies™, A14527) with ExpiFectamine™ 293 (ThermoFisher, A14525) as per the manufacturer's guidelines. Transfected cells were incubated for 4 days in an orbital shaker at 37 °C, 5% CO$_2$, 70% humidity with 1000 rpm shaking, followed by cell harvesting and purification of the Fab from supernatants using anti-C$_H$1 resin (Thermo Scientific, 194320010). The supernatant was incubated overnight with 100 µL of anti-C$_H$1 resin diluted 5-fold in PBS and afterward transferred to a Unifilter membrane (GE Healthcare, 7700-2804) and loaded onto a 96 deep well microplate. In centrifugation intervals of 1 min, $1000 \times g$, the flowthrough was removed, then the resin was washed twice with 500 µL PBS, and the Fab was eluted with 75 µL, 0.2 M Glycine, pH 2.6 into a new plate containing 25 µL of 2 M Tris pH 8.0 neutralization buffer. Finally, the Fab was desalted using Zeba Spin Desalting plates (Thermo Scientific, 89808) into PBS.

The B12 Fab was expressed by transfection of ExpiCHO cells, which were cultured and transfected according to the manufacturer's guidelines (Gibco™). The protocol included addition of ExpiFectamine™ CHO Enhancer and a single feed at Day 1 and cells were maintained at 37 °C and 5% CO$_2$. The supernatant was collected at Day 7 by removal of the cells through centrifugation at 300 g for 5 min, followed by an additional centrifugation at 1000 g for 5 min and storage at −80 °C. After thawing overnight at 4 °C, the supernatant was centrifuged once more, and the His-tagged protein was purified by immobilized metal ion affinity chromatography employing a 5-mL HisTrap Excel column (Cytiva). Fractions of interest were pooled, and the volume reduced by centrifugal filtration using an Amicon Ultra-15 centrifugal filter unit (10 kDa NMWL). The protein solution was subjected to size-exclusion chromatography by injection on a HiLoad 16/600 Superdex75 column (Cytiva) using Dulbecco's PBS as eluent. Fractions of interest were pooled, sterile-filtered, concentrated and subjected to endotoxin-removal using Pierce High-capacity endotoxin removal spin columns. The final concentration was determined by measuring the absorbance at 280 nm on a Nanodrop2000 instrument. Purity was checked by SDS-PAGE. The purified protein was stored at −80 °C.

### Conversion from scFv to IgG format and IgG production
Antibody variable domains were converted from scFv to IgG format prior to expression in CHO-S cells. The variable chains (V$_L$ and V$_H$) were extracted from the pSANG10-3F vector by PCR and were cloned into a single expression vector using the NEBuilder® cloning technique. The expression vector contained the constant domain sequences of the respective human IgG heavy chain (with LALA[23] mutation, or LALA and YTE[24] mutations) and human lambda light chain. For production of IgG (LALA + YTE), a CHO-S cell line with pre-established landing pad suitable for recombinase-mediated cassette exchange with the IgG expression

vector was cultivated in CD CHO medium (Gibco), supplemented with 8 mM L-Glutamine (Thermo Fisher Scientific) and 2 µL/mL anti-clumping agent (Gibco) at 37 °C, 5% CO$_2$ at 120 rpm (shaking diameter 25 mm). The cell line was transfected with IgG expression vector and Cre-recombinase vector in 3:1 ratio (w:w) in 6-well plates (BD Biosciences) at a concentration of $10^6$ cells/mL using FreeStyle MAX transfection reagent (Thermo Fischer Scientific) according to the manufacturer's recommendation. Stable cell pools were generated by adding 5 µg/mL blasticidin five days post-transfection, continuing until cell death of untransfected cells and complete recovery of transfected cells (>95% viability). IgG producing cell pools were seeded at $3 \times 10^5$ cells/mL in FortiCHO or CD CHO medium supplemented with 8 mM glutamine and 2 µL/mL anti-clumping reagent. The pools were cultivated in batch mode (CD CHO) for 5 days or in fed-batch mode (FortiCHO) for 7 to maximum 13 days, feeding glutamine, glucose, and feeding supplements cell boost 7a and 7b (VWR) starting on either day 3, 4, or 5. Cultures were harvested by centrifuging the cell suspension at $300 \times g$ for 10 min followed by $4700 \times g$ for 15 min, removing cells and cell debris. Clarified supernatants were stored at −80 °C until purification. The production of IgG (LALA) was carried out using transient transfection of ExpiCHO cells as described for the B12 Fab above.

### Purification of IgGs
Supernatant was thawed overnight at 4 °C, centrifuged, filtered, and loaded on a MabSelect column (Cytiva). 20 mM sodium phosphate and 150 mM NaCl (pH 7.2) was used for equilibration, and washing of the column and elution was performed with 0.1 M sodium citrate (pH 3). Elution fractions were immediately neutralized by 1 M Tris (pH 9) using 1/5 volume of neutralization solution for 1 volume of elution fraction. Fractions of interest were pooled and loaded on a HiPrep 26/10 desalting column for buffer exchange to Dulbecco's PBS. Protein fractions were sterile-filtered and concentrated by centrifugal filtration using an Amicon® Ultra-15 centrifugal filter unit (30 kDa NMWL). The final concentration was determined by measuring the absorbance at 280 nm on a Nanodrop2000 instrument. Purity was checked by SDS-PAGE. The purified protein was stored at −80 °C.

### Monoclonal IgG ELISA
A monoclonal IgG dose-response ELISA was set up to test whether the binding properties of the scFv were retained after reformatting to the IgG1 format. Biotinylated myotoxin II (5 µg/mL) was captured in streptavidin-coated (10 µg/mL) MaxiSorp plates (Nunc) and blocked with 3% M-PBS. Next, the antibodies were serial diluted (6.85 nM, 3.42 nM, 1.71 nM, 0.85 nM, 0.43 nM, and 0.21 nM) in 3% M-PBS and added in triplicates to the plates. For detection of binding, a 1:10,000 dilution of Anti-Human IgG (Fc-specific)-Peroxidase antibody (Sigma-Aldrich, A0170) in 3% (w/v) skimmed milk in PBS was utilized along with a 3,3',5,5'-Tetramethylbenzidine (TMB) substrate kit (Thermo Scientific, #34021). Absorbance was measured at 450 nm in a Multiskan FC Microplate Photometer (Thermo Scientific). Prior to carrying out the IgG ELISAs, the antibody concentration in the supernatants was determined using an Octet® R8 (Sartorius) with proteinA biosensors. Sensors were hydrated and neutralized in 1x PBS while sensors were regenerated using a 10 mM glycine pH 1.7 buffer. Incubation lasted for 5 min with 1000 rpm shaking speed. The standard curve covered concentrations from 0.78 to 200 µg/mL and was diluted from a concentrated (10 mg/mL) stock of the antibody rituximab that was produced in-house.

### Human fibroblast assay
An in vitro neutralization assay to assess cell viability was carried out using human dermal fibroblasts (neonatal; 106-05 N, Sigma Aldrich) based on the CellTiter-Glo® Luminescent Cell Viability Assay (Cat.# G7571, Promega, USA). First, the IC$_{50}$ of myotoxin II was determined on fibroblasts (29 µg/mL) and subsequently IgGs were assessed for their

neutralizing abilities against 2.5 $IC_{50}$s of myotoxin II (72.5 µg/mL). The level of neutralization was tested at toxin:IgG molar ratios of 1:0.5, 1:1, and 1:1.5.

The assay was carried out according to the manufacturer's protocol over a span of three consecutive days and all experiments were run in triplicates: On the first day, 100 µL of cells (400,000 cells/ mL in fibroblast culture medium) were seeded in 96-well plates and grown for 24 h at 37 °C/5% $CO_2$. After 24 h of incubation, the toxin, or toxin-IgGs mixtures (preincubated 30 min at 37 °C) were added to their respective wells. Following 24 h of incubation (37 °C/5% $CO_2$), the wells were emptied and 100 µL CellTiter-Glo® Reagent (room temperature) was added to each well and the plates were incubated 5–10 min on an orbital shaker. After 10 min at room temperature, the luminescence was recorded (Victor Nivo, Perkin Elmer).

## Bio-layer interferometry measurement
1x Kinetic Buffer (KB, ForteBio) prepared in PBS was used as the running buffer in the bio-layer interferometry experiments (Octet, Sartorious). Prior to the assay, the streptavidin (SAX) biosensors were pre-wet for 10 min in 1x KB. Kinetic assays were performed by first capturing biotinylated myotoxin II using SAX biosensors followed by a 120 s baseline step in 1x KB. The myotoxin II-captured biosensors were then dipped in wells containing increasing concentrations of Fab or IgG – 0 nM, 1 nM, 10 nM, and 100 nM for a 600 s of association step, followed by a dissociation step in 1x KB for 600 s. The experiment was performed at 30 °C with a shaking at 1000 rpm. ForteBio's data analysis software was used to fit the binding curves using a 1:1 binding model to determine the $k_{on}$, $k_{off}$, and $K_D$ except for G08, which was fitted using a 2:1 heterogeneous ligand model, which assumes two sets of rate constants. This 2:1 heterogeneous ligand model was picked because of the biphasic nature of the association curves, where an initial fast on-rate was followed by a slower on-rate, instead of reaching an equilibrium.

## Animals
Animal experiments were conducted using CD-1 mice of both sexes weighing 18–20 g (corresponding to 3–4 weeks old). Mice were supplied by Instituto Clodomiro Picado and experiments were conducted following protocols approved by the Institutional Committee for the Use and Care of Animals (CICUA), University of Costa Rica (approval number CICUA 84–17). Mice were provided food and water *ad libitum* and housed in Tecniplast Eurostandard Type II 1264C cages (L25.0 cm × W40.0 cm × H14.0 cm) in groups of 5 mice per cage. Animals were maintained at 18–24 °C, 60–65% relative humidity and 12:12 light-dark cycle.

## In vivo mouse assay for myotoxicity neutralization
Myotoxin II was preincubated for 30 min at room temperature with either (a) phosphate-buffered saline (0.12 M NaCl, 0.04 M sodium phosphate; PBS, pH 7.2), (b) monoclonal antibody (B12, E02, and G08) at 1:1 molar ratio, or (c) polyvalent equine antivenom (batch 6720721, Instituto Clodomiro Picado) at 1.6 mg toxin/mL antivenom ratio. Subsequently, the preincubated mixtures were injected by intramuscular (gastrocnemius) route, in a total volume of 100 µL (containing 75 µg of toxin as challenge dose), in groups of five CD-1 mice. As a control, a group of mice received an identical injection of 100 µL of PBS alone. Monoclonal antibody alone, in equal amount as in the myotoxin-preincubated mixture, was injected in a group of 5 mice as an additional control. All mice were bled 3 h after injection and the plasma creatine kinase (CK) activity was determined using a UV-kinetic commercial assay (CK-Nac, Biocon Diagnostik), as an indicator of skeletal muscle necrosis[43].

Using the same mouse assay, it was evaluated whether monoclonal antibody B12 (153 µg), B12 Fab format (102 µg), polyvalent equine antivenom (batch 6720721, Instituto Clodomiro Picado) (29 µL), or polyvalent equine antivenom (29 µL) mixed with monoclonal antibody B12 (125 µg), could neutralize the myotoxic effects of whole *B. asper* venom. Venom and antibodies were preincubated for 30 min at room temperature, and then the mixtures were intramuscularly injected in groups of 5 mice (100 µL, containing 50 µg of whole venom as challenge dose), using as a control group of mice receiving PBS alone. Plasma CK activity after 3 h was determined as above.

Mouse experiments followed ethical guidelines of the Institutional Committee for the Use and Care of Animals (CICUA, #084-17) of the University of Costa Rica. Testing of statistical significance was carried out on preselected groups using one-way ANOVA followed by Šídák's multiple comparison test.

## In vivo mouse rescue assays for myotoxicity neutralization
A rescue assay using B12, polyvalent equine antivenom (batch 6720721, Instituto Clodomiro Picado) or polyvalent equine antivenom mixed with monoclonal antibody B12 was performed. *B. asper* venom (50 µg in 100 µL of PBS) was injected in the right gastrocnemius of groups of 5 mice. After 3 min, mice were intravenously injected with either B12 (1085 µg), polyvalent equine antivenom (200 µL), or polyvalent equine antivenom (150 µL) mixed with monoclonal antibody B12 (760 µg). A control group was injected intramuscularly with *B. asper* venom (50 µg in 100 µL of PBS) with no rescue injection after 3 min. Two other control groups were injected intramuscularly with PBS (100 µL), one group received an intravenous injection of B12 (1085 µg) and the other group did not receive a rescue injection. Plasma CK activity after 3 h was determined in all mice groups as above.

Another rescue assay was performed using the same methodology as above, in groups of 5 mice, but instead using the Fab format of B12 or different scaffold mutations for the B12 IgG1, meaning that the scaffold in all above experiments contained both LALA and YTE mutations, but in this experiment, it only contained the LALA mutation. *B. asper* venom (50 µg in 100 µL of PBS) was injected in the right gastrocnemius of groups of 5 mice. After 3 min, mice were intravenously injected with B12(LALA) (985 µg), B12 Fab (664 µg), polyvalent equine antivenom (batch 6720721, Instituto Clodomiro Picado) (200 µL), or polyvalent equine antivenom (150 µL) mixed with B12(LALA) (690 µg). A control group was injected intramuscularly with *B. asper* venom (50 µg in 100 µL of PBS) with no rescue injection after 3 min. A second control group was injected intramuscularly with PBS (100 µL) and did not receive a rescue injection. A third control group was injected intramuscularly with PBS (100 µL), and after 3 min, mice were intravenously injected with B12 Fab (664 µg). Plasma CK activity after 3 h was determined in all mice groups as described previously.

## Histological evaluation of systemic toxicity
Heart, liver, kidney, and lung samples were obtained from groups of mice after 24 h of injection with either *B. asper* venom i.m. (50 µg in 100 µL of PBS) or *B. asper* venom i.m. (50 µg in 100 µL of PBS) followed by i.v. B12(LALA + YTE) (1085 µg) after 3 min. Tissues were fixed in 3.7% formalin, and processed for hematoxylin–eosin staining of paraffin-embedded sections.

## Statistics and reproducibility
No statistical method was used to predetermine the sample sizes. The experiments were not randomized. The investigators were not blinded to allocation during experiments and outcome assessment. In the data analysis of muscle damage measurements in mice, one outlier data point was identified and removed. This data point exhibited a value over three times higher than the mean of the remaining four replicates.

## Reporting summary
Further information on research design is available in the Nature Portfolio Reporting Summary linked to this article.

## Data availability

Protein sequences for the antibodies (in scFv format), TPL0039_05_E02, TPL0039_05_B12, TPL0039_05_F04, TPL0039_05_G08, TPL0039_05_B04, and TPL0039_05_A03, are available in Table S1. The data generated in this study are provided in the Source Data file. Source data are provided with this paper.

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

## Acknowledgements

This research was funded by the Villum Foundation (00025302, A.H.L.), the European Research Council (ERC) under the European Union's Horizon 2020 research and innovation programme (850974, A.H.L.), Wellcome (221702/Z/20/Z, A.H.L.), the Novo Nordisk Foundation (NNF20SA0066621, L.M.G., B.G.V.), and The National Council for Scientific and Technological Development (307184/2020-0, M.B.P.). The authors would like to thank Karoline Schousbou Fremming, Daniel Duun, and Karen Kathrine Brøndum for technical support during antibody purification.

## Author contributions

Conceptualization: C.V.S., A.H.L., Methodology: C.V.S., J.F., A.C.A., H.H.K.W., S.S., L.L., M.B.P., F.A.C., T.T., L.M.G., J.P.M., B.L., A.H.L., Investigation: C.V.S., J.F., A.C.A., H.H.K.W., S.S., L.L., M.B.P., F.A.C., T.T., B.L., Visualization: C.V.S., Funding acquisition: A.H.L., L.M.G., B.G.V., M.B.P., Project administration: C.V.S., A.H.L., Resources: C.V.S., J.F., A.C.A., S.S., L.L., M.F., F.A.C., A.L., B.L., L.M.G., B.G.V., A.H.L., Supervision: A.H.L., B.L., Writing – original draft: C.V.S., J.F., B.L., A.H.L., Writing – review & editing: C.V.S., J.F., A.C.A., H.H.K.W., S.S., L.L., M.B.P., T.T., A.L., A.K.V., J.P.M., B.L., A.H.L.

## Competing interests

The authors declare no competing interests.
