## [Peer Review File · Nature Communications]

Antibody-dependent enhancement of toxicity of myotoxin II from *Bothrops asper*REVIEWER COMMENTS

Reviewer #1 (Remarks to the Author):

The authors describe the development of human antibodies against myotoxin II from *Bothrops asper*. They showed a straightforward antibody generation and analysis project showing a complete story from antibody generation over biochemical analysis, in vitro experiments and in vivo experiments. Most important is the observation of ADE in the in vivo studies!

In the in vivo neutralization (preincubation of venom and B12/anti-venom before i.v.), the B12 alone or the anti-venom is partially protective but the combination of both is fully protective (measured by the creatine level). This approach led to an improved protection against the venom.

Very new and of high interest for the development of recombinant against toxins is the occurrence of ADE in the rescue in vivo experiments (first venom i.m. followed by B12 antibody or anti-venom or both i.v.). Here, the B12 antibody with LALA (reduced Fc-Gamma binding= reduced effector functions) and YTE (improved binding to neonatal receptor = longer half-life) increased disease symptoms (measured by creatine levels). This is unexpected! Because, the venom and the anti-venomen/B12 are not given in the same compartment (i.m., i.v.) a partial toxicity/partial protection like for the anti-venom is expected. With a variant without YTE mutation, the anti-venom is as good as the combination of anti-venom and B12 resulting in a partial protection. The partial protection also differs between both experiments - with/without YTE mutation - for the anti-venom only approach showing the variability of animal experiments. The IgG recycling seems to have an impact on the toxicity of myotoxin II. Interestingly, the corresponding Fab (no recycling, no effector functions) results in a partial protection when giving together with the toxin, but in lower protection in the rescue experiments.

The observation of ADE in the rescue experiments are very important, because the WHO currently recommended preincubation tests for anti-venoms. This WHO recommendation should be revised and rescue/challenge tests should be recommended.

Major Revisions:

-

Minor Revisions:

- I suggest to improve the figures 4-7. The figures are only in gray scale or black, I suggest to include a color code for e.g. myotoxin, venom, B12... and striped bars with the respective colors for the combinations. This would make it easier to understand the graphs.

- I also suggest to add the info "LALA + TYE" above the left graph and "LALA" above the right graph in figure 6.

- Very interesting would be a third graph for figure 6 using IgGs without any mutations (normal effector functions). Will this also lead to ADE?

Reviewer #2 (Remarks to the Author):

The authors here describe a study reporting a set of antibodies selected by phage display screening against myotoxin present in snakebite venoms. Developing rapid snakebite antivenoms is an important global health problem, and several of these antibodies provide some level of protection against myotoxin in vivo and in vitro. However, the authors

surprisingly found that a half-life extension mutation (YTE) actually enhances toxicity in a mouse model. This is an important finding of high relevance to the field.

Overall the study is of high quality. I am not sure that the authors here are observing “classical” ADE like has been observed for viral diseases like dengue. Instead, it appears that the YTE mutation likely traffics the toxin to concentrate near specific cells and/or extend its half life. The authors should be more forceful in separating the enhancement mechanisms for the antivenom YTE antibody from other types of ADE. While the mechanism is important and of value for the field, it is not without precedent and the authors should explore those precedents more fully than is described in the current version of the manuscript. I am not sure that the term ADE applies, but if the authors do choose to use the term ADE, it is critical to fully separate the mechanisms at play here from the more widely studied mechanisms observed in certain viral diseases.

In general, I found this to be a very interesting study in need of moderate revisions. Major comments listed below.

Myotoxin II and other myotoxic PLA2s are responsible for serious 46 clinical effects, such as tissue loss and amputation leading to permanent disability (7).
->What is the known or hypothesized mechanism that leads to tissue loss and amputation? This information would be very helpful for the reader.

PMID: 3188055 (ref 24) suggests that enhanced toxicity of antibody conjugated in Fab formats results from glomerular filtration and tubular reabsorption of the Fab-toxin complex and, to a lesser extent, of the immunoglobulin-toxin complex, which is still toxic to some degree. Have the authors fully explored this mechanism for toxicity in the present model? If not, the authors should show data that demonstrates that the mechanism of toxicity in the kidney is not at play in the present model.
->To this point – Fig. S3 appears to show changes in the kidney, but not substantial changes in other organs.
->Is it possible that the formation of toxic immune complexes and associated cell debris enhances trafficking to the kidney and associated kidney damage in the B12 YTE rescue model? If there was some way to test or explore this hypothesis, it would dramatically strengthen the manuscript.
->The removal of YTE eliminating the effect strongly implicates FcRn trafficking as part of the mechanism for damage (which is expressed on various tissues including kidney and endothelial cells)

Several of the figures for in vivo experiments do not clearly delineate the full molecule used for those experiments (e.g. IgG+LALA+YTE, Fab, etc.) Each figure should indicate the exact molecule used for in vivo studies, fully specified for each condition. Indication only “Venom + B12” as in Fig. 5 does not fully specify the experiment and the different formats quickly get confusing. Ideally, the format would be specified directly in the figure (not just described in the figure legend), especially if multiple antibody formats are used in the same figure.

How quickly does myotoxin clear the body on its own? Is it possible that the YTE mutation

extends the amount of time the toxin circulates, due to the enhanced half-life of the toxin-protein complex?

It is important to note that ADE observed in vitro does not necessarily translate into clinically enhanced disease – even when ADE is observed in mouse models. SARS-CoV-2 has demonstrated the importance of this point. Many in vitro models have shown ADE for SARS-CoV-2 (e.g. in the Syrian hamster model), whereas ADE has never been demonstrated to play a statistically significant/quantifiable role for SARS-CoV-2 therapeutics and vaccines. The role of ADE in natural clinical progression is uncertain – but even those individuals with fading antibody titers after vaccination (or also in the late phase of mAb treatments, although those data are more limited) do not appear to show worse clinical outcomes after SARS-CoV-2 infection despite the potential threat of ADE raised by in vivo model experiments. Thus, tremendous caution is merited for translating in vitro results to clinical impact. The authors really need to discuss the major differences between clinical ADE and in vitro ADE in extensive detail in the discussion, and emphasize that the carefully controlled in vivo mouse models might still not actually translate into clinically significant ADE in human settings.

Due to the importance of the above point, the abstract should also be modified to reflect current understanding and lack of human ADE demonstrated. One suggestion: “While clinical ADE related to snakebite venom has not yet been reported in humans, this first report of ADE of a toxin from the animal kingdom highlights the necessity of assessing even well-known antibody formats in representative preclinical models to evaluate their therapeutic utility against toxins or venoms, to avoid potential deleterious effects as exemplified in the present in vivo model.”

The abstract must also include a statement that the ADE observed in vivo was dependent on the use of a half-life extension mutation (YTE), and that not ADE was observed using standard antibody IgG or Fab formats. This information is critical to understand and interpret study results, and thus should be included in abstract.

The sequences of the antibody clones analyzed in detail (TPL0039_05_E02, 111 TPL0039_05_B12, TPL0039_05_F04, TPL0039_05_G08, TPL0039_05_B04, and 112 TPL0039_05_A03) should be deposited in a publicly accessible repository prior to publication.

In this assay, three molar ratios of toxin:IgG were tested;
131 1:0.5, 1:1, and 1:1.5.

->Is the toxin a monomer? Trying to understand how the number of binding sites match up between toxin and IgG in this assay, it would be helpful to clarify here for the reader

Fig. 3 - TPL0039_05_E02 and TPL0039_05_F04 IgGs are toxic to cells at some level.
Would be helpful to clearly highlight in the results.

The ADE phenomenon is also known from other fields,
295 such as vaccinology, where it is connected with viral enhancement (21, 22). Recently,
ADE was
296 identified as the underlying cause of breakthrough severe dengue disease in children
who had not
297 been previously infected with dengue but had received a yellow fever chimeric
tetravalent dengue
298 vaccine (28). These examples highlight the importance of investigating and
understanding ADE,
299 as this effect can be detrimental for new therapeutics.

->The authors need to be very careful to draw parallels like this. The mechanisms for ADE in
dengue are completely different from the mechanisms here. I suggest that the authors
specify the mechanism of ADE in dengue, and note that an entirely different phenomenon
must be at play for a myotoxin that does not preferentially replicate in immune cells like
dengue virus does.

These experiments suggest
310 that the mechanism responsible for the ADE effect might involve the pharmacokinetic
properties
311 of the TPL0039_05_B12 monoclonal antibody.
->Need to specify more clearly on line 311 – “pharmacokinetic properties of the
TPL0039_05_B12 YTE-modified monoclonal antibody”. TPL0039_05_B12 IgG shows no
ADE.

we carried out histological observations of the kidney, liver, lung, and heart sections from
326 the mice that died after TPL0039_05_B12 IgG treatment and compared these to mice
injected with
327 venom alone. However, these studies failed to provide clues on the toxic effects
involved. As such,
328 the basis of this relevant ADE mechanism remains largely unknown.

->I see major differences in the kidney tissue morphology in Fig. S3 (panel e vs. panel f). Do
the authors agree?

our findings highlight a previously unidentified limitation of
332 preincubation assays, which are currently used as the gold standard for assessing the
preclinical
333 efficacy of toxin-neutralizing agents in the field of antivenom research (33)
->Completely agree! This is a very important finding for the field

I believe that the authors are not showing ADE as it is normally defined. Rather, it appears that the authors are showing that half-life enhancing mutations for anti-toxin antibodies actually increases the amount of time a toxin remains exposed to circulation, and potentially concentrates antibody at FcRn-expressing cells (such as in the kidney), thereby exacerbating disease. A similar phenomenon has been observed in antibody-drug conjugates (ADC's) with highly toxic conjugates. The authors may be able to find appropriate citations to draw parallels to ADC's.

The authors should add a note to the discussion that antivenoms may not benefit from FcRn-mediated half-life extension mutations, for two reasons. 1) Enhanced FcRn affinity of half-life extension mutations could enhance toxin trafficking to cells that express FcRn, such as kidney and epithelial cells, and 2) Half-life extension mutations could extend the residence time for toxins in the body, enhancing exposure. Thus, antivenom antibodies may be more effective as unmodified IgG, or even potentially other formats, to avoid these issues.

581 Authors declare that they have no competing interests.

->Have any patents been filed on any of the antibodies in the study? Or will a patent be filed on any of the discovered antibodies prior to public release of the antibody sequences with this publication?

Reviewer #3 (Remarks to the Author):

The manuscript titled "Antibody-dependent enhancement of toxicity of myotoxin II from *Bothrops asper*" by Sørensen et al. demonstrates that a human monoclonal antibody results in antibody-dependent enhancement of the toxic effects exerted by myotoxin II from *Bothrops asper* in an in-vivo rescue mice model. Such antibody-dependent enhancement of the toxic effect has not been previously observed for an animal toxin group.

The manuscript contains findings significant to the field of concern, and is well-written. The experiments are appropriate and are clearly explained.

In-vivo rescue model: The time gap between intramuscular venom injection and intravenous antivenom or antivenom+MAb is 3 min. Why this time gap is as narrow as 3 min, despite venom was given im and the antivenom in iv? Is there a justification for that?

Figure 3, values have been normalised to the negative control and Figures 4-7: values have been normalised to the positive control. I understand the authors have done this to achieve the graph consistency. However, I think the reader would get a more clear picture of the effects of all treatment groups if these were expressed realistically, rather than normalising to positive or negative control.

REVIEWER COMMENTS

Reviewer #1 (Remarks to the Author):

The authors describe the development of human antibodies against myotoxin II from *Bothrops asper*. They showed a straightforward antibody generation and analysis project showing a complete story from antibody generation over biochemical analysis, in vitro experiments and in vivo experiments. Most important is the observation of ADE in the in vivo studies! In the in vivo neutralization (preincubation of venom and B12/anti-venom before i.v.), the B12 alone or the anti-venom is partially protective but the combination of both is fully protective (measured by the creatine level). This approach led to an improved protection against the venom. Very new and of high interest for the development of recombinant against toxins is the occurrence of ADE in the rescue in vivo experiments (first venom i.m. followed by B12 antibody or anti-venom or both i.v.). Here, the B12 antibody with LALA (reduced Fc-Gamma binding= reduced effector functions) and YTE (improved binding to neonatal receptor = longer half-life) increased disease symptoms (measured by creatine levels). This is unexpected! Because, the venom and the anti-venom/B12 are not given in the same compartment (i.m., i.v.) a partial toxicity/partial protection like for the anti-venom is expected. With a variant without YTE mutation, the anti-venom is as good as the combination of anti-venom and B12 resulting in a partial protection. The partial protection also differs between both experiments - with/without YTE mutation - for the anti-venom only approach showing the variability of animal experiments. The IgG recycling seems to have an impact on the toxicity of myotoxin II. Interestingly, the corresponding Fab (no recycling, no effector functions) results in a partial protection when giving together with the toxin, but in lower protection in the rescue experiments. The observation of ADE in the rescue experiments are very important, because the WHO currently recommended preincubation tests for anti-venoms. This WHO recommendation should be revised and rescue/challenge tests should be recommended.

Overall response: The authors thank the reviewer for the comments. We believe they have improved the manuscript, and especially the comments about the figures have made the manuscript more interesting to read.

Major Revisions:

-

Minor Revisions:

- I suggest to improve the figures 4-7. The figures are only in gray scale or black, I suggest to include a color code for e.g. myotoxin, venom, B12... and striped bars with the respective colors for the combinations. This would make it easier to understand the graphs.

Answer: The authors thank the reviewer for this comment to improve the figures with colors and patterns. We have changed figures 2-7 and made the colors consistent between these, which should make it easier to follow the specific antibodies through the figures.

- I also suggest to add the info "LALA + TYE" above the left graph and "LALA" above the right graph in figure 6.

Answer: The authors thank the reviewer for this comment. Based on this comment and a comment from reviewer 2, we have changed the antibody names in all neutralization figures (Fig. 3-7) to include which Fc mutations the antibodies contain. Additionally, we have changed all appearances of an antibody name to include its Fc mutations or formats i.e. B12(LALA), B12(LALA+YTE), B12(Fab).

- Very interesting would be a third graph for figure 6 using IgGs without any mutations (normal effector functions). Will this also lead to ADE?

Answer: We thank the reviewers for this comment. After deliberation, the authors do not believe that the sacrifice of additional animals is warranted as it is highly unlikely that the results will differ from those seen for the IgG(LALA), since this mutation is only there to decrease Fc effector functions, as well as ADE is also seen for the Fab format, which does not have any effector function.

Reviewer #2 (Remarks to the Author):

The authors here describe a study reporting a set of antibodies selected by phage display screening against myotoxin present in snakebite venoms. Developing rapid snakebite antivenoms is an important global health problem, and several of these antibodies provide some level of protection against myotoxin in vivo and in vitro. However, the authors surprisingly found that a half-life extension mutation (YTE) actually enhances toxicity in a mouse model. This is an important finding of high relevance to the field.

Overall the study is of high quality. I am not sure that the authors here are observing “classical” ADE like has been observed for viral diseases like dengue. Instead, it appears that the YTE mutation likely traffics the toxin to concentrate near specific cells and/or extend its half life. The authors should be more forceful in separating the enhancement mechanisms for the antivenom YTE antibody from other types of ADE. While the mechanism is important and of value for the field, it is not without precedent and the authors should explore those precedents more fully than is described in the current version of the manuscript. I am not sure that the term ADE applies, but if the authors do choose to use the term ADE, it is critical to fully separate the mechanisms at play here from the more widely studied mechanisms observed in certain viral diseases.

In general, I found this to be a very interesting study in need of moderate revisions. Major comments listed below.

Overall response: The authors thank the reviewer for these elaborate comments and requests. We have implemented almost all suggestions and believe it has improved the manuscript significantly. We have tried to separate the ADE label in our study from the ADE label in viral enhancement studies, as we think that ADE is the right terminology to use (as long as we specify that ADE of a toxin is very different from ADE of a virus particle). However, if the reviewer has a better suggestion, we are very open to suggestions.

One suggestion from us could be to call this phenomenon ADET (antibody-dependent enhancement of toxicity), although we do think that ADE should signify a more general

phenomenon and that ADE should always be specified in relation to the pathogenic agent (which theoretically might also be bacteria, parasites etc.)

Myotoxin II and other myotoxic PLA2s are responsible for serious 46 clinical effects, such as tissue loss and amputation leading to permanent disability (7).
->What is the known or hypothesized mechanism that leads to tissue loss and amputation? This information would be very helpful for the reader.

Answer: The authors thank the reviewer for this comment and agree that this would be very helpful for the reader. We have added added information of this to the section XXX-XXX.

PMID: 3188055 (ref 24) suggests that enhanced toxicity of antibody conjugated in Fab formats results from glomerular filtration and tubular reabsorption of the Fab-toxin complex and, to a lesser extent, of the immunoglobulin-toxin complex, which is still toxic to some degree. Have the authors fully explored this mechanism for toxicity in the present model? If not, the authors should show data that demonstrates that the mechanism of toxicity in the kidney is not at play in the present model.

->To this point – Fig. S3 appears to show changes in the kidney, but not substantial changes in other organs.

Answer: We highly appreciate the reviewer for pointing this out. Upon the reviewer's feedback we went back to the saved tissue slides and found that we had indeed missed some kidney damage, which gives important clues for the underlying mechanism. Therefore, we have now removed the original kidney pictures from Fig. S3 and instead constructed a new figure with additional histological images focusing on the kidney damage and implemented this as a figure in the main manuscript (Fig. 7). Further, we have changed the description of the histological observations in different places, including lines 256-268 and 278-285.

->Is it possible that the formation of toxic immune complexes and associated cell debris enhances trafficking to the kidney and associated kidney damage in the B12 YTE rescue model? If there was some way to test or explore this hypothesis, it would dramatically strengthen the manuscript.

Answer: We thank the reviewer for this good question. It is indeed something the authors have been wondering as well. The authors are unsure if the observed kidney damage is due to a similar phenomenon as described by the reviewer or is simply due to increased muscle damage, or something else (which the results from the Fab version of the antibody might indicate, see next reviewer comment and response). In line with our response above, we have provided more details and histological images of the kidney damage, and re-written parts of the results and discussion (lines 256-268 and 278-285).

->The removal of YTE eliminating the effect strongly implicates FcRn trafficking as part of the mechanism for damage (which is expressed on various tissues including kidney and endothelial cells)

Answer: The authors agree, however, since the Fab format does not cause increased myotoxicity but still results in fatal outcomes for the mice, it seems that something else is at play as well. To emphasize that the FcRn trafficking could play a role we added this to line 355-357

“These results indicate that the toxicity could be related to an increased half-life of the antibody-toxin complex or to the increased FcRn-mediated uptake of the antibody-toxin complex.”

Several of the figures for in vivo experiments do not clearly delineate the full molecule used for those experiments (e.g. IgG+LALA+YTE, Fab, etc.) Each figure should indicate the exact molecule used for in vivo studies, fully specified for each condition. Indication only “Venom + B12” as in Fig. 5 does not fully specify the experiment and the different formats quickly get confusing. Ideally, the format would be specified directly in the figure (not just described in the figure legend), especially if multiple antibody formats are used in the same figure.

Answer: The authors thank the reviewer for this comment and have added the Fc mutations directly in the graph in all neutralization figures (Fig 3-7). Additionally, we have changed all appearances of an antibody name to include its Fc mutations or formats i.e. B12(LALA), B12(LALA+YTE), B12(Fab).

How quickly does myotoxin clear the body on its own? Is it possible that the YTE mutation extends the amount of time the toxin circulates, due to the enhanced half-life of the toxin-protein complex?

Answer: The information about myotoxin II half-life is to the best of our knowledge unfortunately not available. However, we also think this enhanced toxicity could be due to the antibody increasing the half-life of the toxin. To emphasize this hypothesis more clearly, we have added this to line 355-357:

“ These results indicate that the toxicity could be related to an increased half-life of the antibody-toxin complex or to the increased FcRn-mediated uptake of the antibody-toxin complex.

It is important to note that ADE observed in vitro does not necessarily translate into clinically enhanced disease – even when ADE is observed in mouse models. SARS-CoV-2 has demonstrated the importance of this point. Many in vitro models have shown ADE for SARS-CoV-2 (e.g. in the Syrian hamster model), whereas ADE has never been demonstrated to play a statistically significant/quantifiable role for SARS-CoV-2 therapeutics and vaccines. The role of ADE in natural clinical progression is uncertain – but even those individuals with fading antibody titers after vaccination (or also in the late phase of mAb treatments, although those data are more limited) do not appear to show worse clinical outcomes after SARS-CoV-2 infection despite the potential threat of ADE raised by in vivo model experiments. Thus, tremendous caution is merited for translating in vitro results to clinical impact. The authors really need to discuss the major differences between clinical ADE and in vitro ADE in extensive detail in the discussion, and emphasize that the carefully controlled in vivo mouse models might still not actually translate into clinically significant ADE in human settings.

Answer: We thank the reviewer for this elaborate comment, it is indeed a good idea to discuss this. To ensure that this is made clear we have added the following to line 343-347

“In this relation, it is important to note, though, that ADE observed in *in vivo* models might not translate into clinically significant ADE in humans, and might also differ based on the in vivo model or the dosing regimen used, as has been observed with vaccine-associated enhanced respiratory

disease in COVID-19 vaccines” and referred to this review:

“Muñoz-Fontela C, Widerspick L, Albrecht RA, Beer M, Carroll MW, de Wit E, Diamond MS, Dowling WE, Funnell SG, García-Sastre A, Gerhards NM. Advances and gaps in SARS-CoV-2 infection models. PLoS pathogens. 2022 Jan 13;18(1):e1010161.”

Due to the importance of the above point, the abstract should also be modified to reflect current understanding and lack of human ADE demonstrated. One suggestion: “While clinical ADE related to snakebite venom has not yet been reported in humans, this first report of ADE of a toxin from the animal kingdom highlights the necessity of assessing even well-known antibody formats in representative preclinical models to evaluate their therapeutic utility against toxins or venoms, to avoid potential deleterious effects as exemplified in the present in vivo model.”

Answer: We have implemented this with slight modification so that part of the abstract now reads like this:

“While clinical ADE related to snake venom has not yet been reported in humans, this first report of ADE of a toxin from the animal kingdom highlights the necessity of assessing even well-known antibody formats in representative preclinical models to evaluate their therapeutic utility against toxins or venoms. This is essential to avoid potential deleterious effects as exemplified in the present study.”

The abstract must also include a statement that the ADE observed in vivo was dependent on the use of a half-life extension mutation (YTE), and that not ADE was observed using standard antibody IgG or Fab formats. This information is critical to understand and interpret study results, and thus should be included in abstract.

Answer: The authors thank the reviewer for this point, however, we would like to highlight, that some form of ADE was also observed for the Fab formats, just not increased myotoxicity, but instead the treatment resulted in fatal outcomes for some mice without showing myotoxicity, whereas only venom treated mice were doing fine. Therefore we do not think we can conclude that the increased toxicity was solely due to the YTE mutation.

The sequences of the antibody clones analyzed in detail (TPL0039_05_E02, 111 TPL0039_05_B12, TPL0039_05_F04, TPL0039_05_G08, TPL0039_05_B04, and 112 TPL0039_05_A03) should be deposited in a publicly accessible repository prior to publication.

Answer: Protein sequences are now accessible in Supplementary Table 1 and the Data and Materials Availability statement changed to: “Protein sequences of the antibodies (scFv formats) TPL0039_05_E02, TPL0039_05_B12, TPL0039_05_F04, TPL0039_05_G08, TPL0039_05_B04, and TPL0039_05_A03 are available in Supplementary Table S1. Other data is available upon request.”

In this assay, three molar ratios of toxin:IgG were tested;
131 1:0.5, 1:1, and 1:1.5.

->Is the toxin a monomer? Trying to understand how the number of binding sites match up between toxin and IgG in this assay, it would be helpful to clarify here for the reader

Answer: We thank the reviewer for this question, we agree that this information would be helpful for the reader. We have added information on this in line 53-55

Fig. 3 - TPL0039_05_E02 and TPL0039_05_F04 IgGs are toxic to cells at some level. Would be helpful to clearly highlight in the results.

Answer: The authors thank the reviewer for this comment and have changed line 155-157 to "The two IgGs, E02(LALA+YTE) and F04(LALA+YTE), showed a lower cell viability than the negative control at all concentrations, with increasing IgG concentrations resulting in decreasing cell viability"

The ADE phenomenon is also known from other fields,
295 such as vaccinology, where it is connected with viral enhancement (21, 22). Recently, ADE was
296 identified as the underlying cause of breakthrough severe dengue disease in children who had
not
297 been previously infected with dengue but had received a yellow fever chimeric tetravalent
dengue
298 vaccine (28). These examples highlight the importance of investigating and understanding
ADE,
299 as this effect can be detrimental for new therapeutics.

->The authors need to be very careful to draw parallels like this. The mechanisms for ADE in dengue are completely different from the mechanisms here. I suggest that the authors specify the mechanism of ADE in dengue, and note that an entirely different phenomenon must be at play for a myotoxin that does not preferentially replicate in immune cells like dengue virus does.

Answer: The authors thank the reviewer for this great comment, it is indeed important to note that completely different mechanisms are at play. To reflect this, we have changed line 334-337 to this:

"While an entirely different mechanism must be at play in ADE of toxins compared to viruses (given the fundamental differences between a toxic protein and a replicating entity such as a virus), the above examples highlight the importance of investigating and understanding ADE, as this effect can be detrimental to new therapeutics."

These experiments suggest

310 that the mechanism responsible for the ADE effect might involve the pharmacokinetic properties

311 of the TPL0039_05_B12 monoclonal antibody.

->Need to specify more clearly on line 311 – "pharmacokinetic properties of the TPL0039_05_B12 YTE-modified monoclonal antibody". TPL0039_05_B12 IgG shows no ADE.

Answer: The authors agree with the reviewer and thank the reviewer for suggesting this

improvement. To also agree with an earlier comment from the reviewer, we have changed the line 346-356 to the below. If this is still not optimal, we are happy to make further changes:

“Interestingly, envenomed mice treated with B12(LALA), in contrast to those treated with B12(LALA+YTE), did not show behavioral alterations or deaths, and did not alter plasma CK levels, as compared to mice injected with venom alone. These results indicate that the toxicity could be related to an increased half-life of the antibody-toxin complex or to increased FcRn-mediated uptake of the antibody-toxin complex. When tested in the Fab format, the B12(Fab) antibody demonstrated significant neutralization of myotoxicity in both the preincubation and rescue *in vivo* assays, while simultaneously causing the death of one out of five mice in each assay. Thus, this Fab antibody experiment indicates that the mechanism responsible for the ADE effect might also involve other pharmacokinetic properties than the YTE-related increased half-life of the B12 monoclonal antibody.”

we carried out histological observations of the kidney, liver, lung, and heart sections from 326 the mice that died after TPL0039_05_B12 IgG treatment and compared these to mice injected with 327 venom alone. However, these studies failed to provide clues on the toxic effects involved. As such, 328 the basis of this relevant ADE mechanism remains largely unknown.

->I see major differences in the kidney tissue morphology in Fig. S3 (panel e vs. panel f). Do the authors agree?

Answer: Again, we want to express our deep gratitude to the reviewer for pointing this out, which we had simply not detected in our initial histological examinations. As mentioned earlier, the authors revisited the tissue slides and agree with the reviewer in observing differences in the kidney morphology. The major implementations the authors have made based on this can be found in Fig. 7, Fig. S3, lines 256-268, and lines 278-285.

our findings highlight a previously unidentified limitation of 332 preincubation assays, which are currently used as the gold standard for assessing the preclinical

333 efficacy of toxin-neutralizing agents in the field of antivenom research (33)

->Completely agree! This is a very important finding for the field

Answer: We are happy that the reviewer agrees with the relevance of this finding

I believe that the authors are not showing ADE as it is normally defined. Rather, it appears that the authors are showing that half-life enhancing mutations for anti-toxin antibodies actually increases the amount of time a toxin remains exposed to circulation, and potentially concentrates antibody at FcRn-expressing cells (such as in the kidney), thereby exacerbating disease. A similar phenomenon has been observed in antibody-drug conjugates (ADC's) with highly toxic conjugates. The authors may be able to find appropriate citations to draw parallels to ADC's.

Answer: The authors thank the reviewer for this comment. However, since the Fab format is also causing ADE (in the form of increased mortality, not increased CK levels), we cannot fully conclude

that this is due only to increased half-life. We know that ADE is normally used in regards to viral enhancement, however, since the enhancement of toxicity only occurred with the addition of the antibody we thought the label ADE was very fitting as well. We are happy to label it otherwise if the reviewer would prefer this (and we would be happy to receive suggestions).

The authors should add a note to the discussion that antivenoms may not benefit from FcRn-mediated half-life extension mutations, for two reasons. 1) Enhanced FcRn affinity of half-life extension mutations could enhance toxin trafficking to cells that express FcRn, such as kidney and epithelial cells, and 2) Half-life extension mutations could extend the residence time for toxins in the body, enhancing exposure. Thus, antivenom antibodies may be more effective as unmodified IgG, or even potentially other formats, to avoid these issues.

Answer: The authors thank the reviewer to bring attention to this, but we do not think we have enough data to support such claims as these could potentially change how future antivenoms are developed. Our study just shows one example of an antibody doing this, with one specific subtype of toxin. From conventional antivenoms, it is known that the Fab format has a problematically short half-life, as some toxins exit the bite site late (due to venom depot effects) and exert toxicity in the absence of Fabs that are already cleared (see e.g. DOI: 10.3109/15563650.2014.974263). So, it might be that some toxins are better neutralized with less antibody material, if the antibody is both half-life extending AND binds the toxin sufficiently tight AND the toxin is inactive when bound to the antibody (the latter is shown to be the case for snake alpha-neurotoxins in several studies, incl. <https://doi.org/10.1038/s41467-023-36393-4>).

It would be very interesting to carry out a study to establish what is the best format for future antivenoms, since the information would be of high value to future antivenom development. However, we find that such a study is beyond the scope of this one

581 Authors declare that they have no competing interests.

->Have any patents been filed on any of the antibodies in the study? Or will a patent be filed on any of the discovered antibodies prior to public release of the antibody sequences with this publication?

Answer: A patent application was once filed but has been discontinued due to the observed issues with ADE of toxicity. The authors have no commercial interest in the work being presented, and the antibody sequences are being made publicly available (Table S1).

Reviewer #3 (Remarks to the Author):

The manuscript titled “Antibody-dependent enhancement of toxicity of myotoxin II from *Bothrops asper*” by Sørensen et al. demonstrates that a human monoclonal antibody results in antibody-dependent enhancement of the toxic effects exerted by myotoxin II from *Bothrops asper* in an in-vivo rescue mice model. Such antibody-dependent enhancement of the toxic effect has not been

previously observed for an animal toxin group.

The manuscript contains findings significant to the field of concern, and is well-written. The experiments are appropriate and are clearly explained.

Overall response: We thank the reviewer for the relevant questions and comments, and we hope we have answered them in a satisfactory way.

In-vivo rescue model: The time gap between intramuscular venom injection and intravenous antivenom or antivenom+MAb is 3 min. Why this time gap is as narrow as 3 min, despite venom was given im and the antivenom in iv? Is there a justification for that?

Answer: This is a very good question that the reviewer asks. Our general reasoning was simply to mimic a very short time delay between envenoming and i.v. treatment, hypothetically considering the preparation or reconstitution of an antivenom. Many papers use a "zero-delay" or immediate i.v. injection (which actually takes not less than 1 min to move the i.m.-injected mouse into a suitable restrainer for i.v. injection), but we thought that 3 min would be better both for such logistic practicality and for considering a more realistic scenario of quick treatment. We also note that 3 min delay in a mouse model would correspond to a significantly longer delay in a human envenoming case due to large differences in pharmacokinetics (but we avoid speculating about this in the manuscript).

Figure 3, values have been normalised to the negative control and Figures 4-7: values have been normalised to the positive control. I understand the authors have done this to achieve the graph consistency. However, I think the reader would get a more clear picture of the effects of all treatment groups if these were expressed realistically, rather than normalising to positive or negative control.

Answer: The authors agree that that would be optimal, however, this would result in figure 4 being confusing. This is because the experiments in figure 4 have been combined from two different days which showed quite different overall signal levels, therefore we would either need to show this in two different graphs or show twice the amount of positive and negative controls in the graph. This is why we picked the normalization method for this graph, as this would allow us to compare these experimental days. Further, to keep consistency throughout the manuscript, we decided to carry out this normalization for all graphs.

REVIEWERS' COMMENTS

Reviewer #1 (Remarks to the Author):

Thanks for the making the changes! I think the paper is fine and can be accepted.

One minor point: in figure 5, 6, 8 and 9, PBS and Venom have the same black and white hatching. I suggest to use a different one.

Reviewer #2 (Remarks to the Author):

I would like to congratulate the authors on a fantastic job addressing comments! I think the manuscript is in great shape.

I very much like the author's suggestion of 'antibody-dependent enhancement of toxicity' (ADET)! I will leave it up to the authors if they want to revise the manuscript further to introduce this term. I believe introducing the term ADET would be tremendously helpful for the field, that would hopefully be cited widely.

Reviewer #3 (Remarks to the Author):

The authors have addressed my comments. I have no further comments.

Point-by-point response to REVIEWERS' COMMENTS

Reviewer #1 (Remarks to the Author):

Thanks for the making the changes! I think the paper is fine and can be accepted.

One minor point: in figure 5, 6, 8 and 9, PBS and Venom have the same black and white hatching. I suggest to use a different one.

Answer: We tried making the two controls different, but we found that it disturbed the graphs and drew attention away from the results we wanted to present. This was in part because the PBS(negative control) bars are so low that the pattern is barely visible in most graphs. We hope you are okay with this and we would like to extend our thanks to you for your help in improving the manuscript.

Reviewer #2 (Remarks to the Author):

I would like to congratulate the authors on a fantastic job addressing comments! I think the manuscript is in great shape.

I very much like the author's suggestion of 'antibody-dependent enhancement of toxicity' (ADET)! I will leave it up to the authors if they want to revise the manuscript further to introduce this term. I believe introducing the term ADET would be tremendously helpful for the field, that would hopefully be cited widely.

Answer: Thank you for the nice words on our manuscript, we really appreciate the feedback we have received from you!

We have implemented the ADET description throughout the manuscript.

Reviewer #3 (Remarks to the Author):

The authors have addressed my comments. I have no further comments.

Answer: Thank you for your earlier comments which improved the manuscript.